# Association between Sociodemographic Factors and Condom Use among Migrant Sex Workers in Chiang Mai, Northern Thailand

**DOI:** 10.3390/ijerph19169830

**Published:** 2022-08-10

**Authors:** Suji Yoo O’Connor, Arunrat Tangmunkongvorakul, Kriengkrai Srithanaviboonchai, Patumrat Sripan, Cathy Banwell, Matthew Kelly

**Affiliations:** 1Department of Global Health, National Centre for Epidemiology and Population Health, Australian National University, Canberra 2601, Australia; 2Research Institute for Health Sciences, Chiang Mai University, 110 Intavaroros Road, Sri-phum, Muang, Chiang Mai 50200, Thailand; 3Faculty of Medicine, Chiang Mai University, Chiang Mai 50200, Thailand; 4Centre for Public Health and Data Policy, National Centre for Epidemiology & Population Health, Australian National University, Canberra 2601, Australia

**Keywords:** sexual behavior, condom use, migrant sex workers, Chiang Mai, Thailand

## Abstract

Thailand has the highest HIV burden in the Asia-Pacific region, with the majority of cases occurring in specific populations. Migrant Sex Workers (MSWs) in Thailand are an important population for HIV risk, yet there has been limited literature on this group and their protective sexual behavior. A cross-sectional study was conducted among 396 MSWs 18–49 years old from 23 sex work-identified venues in Chiang Mai. Participants were surveyed on their own sociodemographic information, health behavior, sexual risk behavior, quality of life, and depression. Male respondents were significantly younger than females (*p* = 0.003). Most respondents were from Myanmar and were ethnic Shan. In the month preceding the survey, 17.0% of MSWs had consistent condom use with regular partners, 53.7% with casual partners, and 87.9% with clients. Condom use was least practiced with regular partners and most practiced with clients (17% and 87.9%, respectively; *p* < 0.001). There was a significant positive association between condom use and starting high school (χ^2^ = 8.08, *p* = 0.018). Education was the only variable that was significantly correlated with condom use with any sexual partner (OR = 0.41; 95%CI 0.20–0.82). Findings of the study indicate that further efforts are needed to promote condom use among migrant sex workers and their sexual partners in Thailand.

## 1. Introduction

As of 2020, there were approximately 5.8 million people living with HIV (PLWHIV), and 240,000 new HIV infections in Asia and the Pacific, the second highest burden after sub-Saharan Africa [1]. Structural barriers such as limited access to testing and healthcare have been cited as important factors in the high HIV burden [2]. This is particularly pertinent given that almost all cases in the Asia-Pacific region occur in key populations, namely, sex workers, men who have sex with men (MSM), transgender people, and people who inject drugs [1,2]. These groups are often marginalized and reluctant to seek healthcare due to structural barriers, such as stigma, fear, and limited education [3]. 

The estimated HIV prevalence in the Asia-Pacific ranged between 0.1% and 1.1% in adult populations as of 2018 [4]. Thailand accounted for the highest adult prevalence (1.1%), followed by Papua New Guinea (0.8%) and Myanmar (0.8%). The lowest prevalence (0.1%) was reported for Australia, Pakistan, Philippines, New Zealand, and Nepal. As of 2020, Thailand had the largest number of PLWHIV in the Asia-Pacific, accounting for 9% of cases in the region [1]. 

A likely factor contributing to the high burden of HIV in Asia-Pacific is inconsistent condom use. It has long been recognized that condom use is a key mechanism in preventing transmission of HIV and other sexually transmitted diseases, yet condom use frequency continues to vary among PLHIV. In the Asia-Pacific, consistent condom use varies between countries [5] where the rate of consistent condom use among PLHIV varies between 27% and 85.3% [5,6]. Consistent condom use was found to be less commonly practiced among sex workers with HIV in the region [5]. This may be partly due to the inequities and social barriers that sex workers face.

Globally, sex workers face higher levels of social inequalities and stigma compared to the general population, which increase their vulnerability to STDs and HIV [5,6,7]. Migrant sex workers (MSWs) are particularly at risk, due to additional barriers such as limited access to services, and marginalization [5,7,8].

## 2. Background

The first case of HIV in Thailand was reported in 1984, but the peak in cases did not occur until the early 1990s, at which point there were over 100,000 new cases annually and HIV/AIDS became an important public health issue [9,10]. Following the surge of HIV cases in Thailand in the early 1990s, several government-led public health programs were implemented [9]. One of the main control strategies that continues to be used is the promotion of condom use in the population. Since 1991, the 100% Condom Use Programme has distributed free condoms and has concentrated on raising awareness and increasing condom availability amongst sex workers [10]. Similar campaigns continued over the following decades. In 2017, the Thai government announced the Thailand National Strategy to End AIDS 2017–2030 [11], with the aim of eradicating AIDS in Thailand. As part of this strategy, the government announced greater efforts to promote condom use and access, with a particular focus on key at-risk populations, where the majority of HIV cases occur. Of all new infections, around 44% occurred among men who have sex with men and 10% among sex workers and their clients [6]. These two groups are also over-represented in terms of prevalence, with around 2% of Thailand’s 147,000 sex workers and 12% of MSM currently infected with HIV [1,12]. However, while studies indicate that HIV cases among MSM and Thai sex workers are declining [13,14], HIV in MSWs in Thailand is increasing [15].

HIV prevalence among MSWs in Thailand is estimated at 5% [16], more than double the estimated prevalence among the general sex worker population [17]. This is of particular importance given the high migrant intake in Thailand, especially from Myanmar and Laos, where populations seek to escape economic hardship and political conflicts at home [18,19]. Chiang Mai is especially affected by these migration patterns due to its proximity to the Thailand–Myanmar border and its status as the metropolitan center and economic hub of northern Thailand. An estimated 200,000 of the 1.2 million population of Chiang Mai are migrants or non-Thai ethnic minorities [20], and this subpopulation comprises up to 50% of sex workers in Northern Thailand [21]. Previous research has found that MSWs in Thailand are particularly at-risk for HIV and other STDs, due to inequities surrounding sociodemographic, access to health messaging and services, and sociopolitical factors [20,22,23]. Unlike Thai sex workers (TSWs), MSWs do not have free STD screening services, meaning that testing costs are out of pocket [24]. Similarly, health messaging for MSWs can be inaccessible, either due to language barriers or lack of opportunity [15,21,24,25]. MSWs may also feel less able to enforce condom use with clients, or may be ignored, increasing the risk of STDs [15,24]. A recent study by Jirritikorn et al. [15] reported instances of MSWs’ clients removing condoms during intercourse without consent [15]. MSWs in Northern Thailand are therefore a key target group for improving condom use and control of HIV/AIDS transmission in the region. It is essential, however, to first understand the patterns of condom use in this at-risk group in order to develop targeted strategies for intervention.

The current study explores patterns of condom use among MSWs in Chiang Mai. The objective of this study was to investigate understandings regarding sociodemographic factors and condom use among MSWs in Chiang Mai, Thailand. Given the reported difficulties for MSWs in enforcing condom use with clients, we paid particular attention to variations in condom use between clients and other sexual partners.

## 3. Materials and Methods

### 3.1. Study Population and Study Design

A cross-sectional design was used. The study population comprised non-Thai sex workers currently living or working in Chiang Mai. The respondents were recruited from diverse sex work-identified venues (i.e., karaoke bars, sauna and spa shops, massage parlors, pubs and restaurants). The eligibility criteria were (1) non-Thai citizens, (2) being at least 18 years of age, and (3) having traded sex for money in the past 12 months.

### 3.2. Ethics Statement

The study received ethical approval from the Human Experimentation Committee, Research Institute for Health Sciences, Chiang Mai University (Certificate of Ethical Clearance No. 4/2019). Prior to data collection, participants were informed about the objectives of the study, risks and benefits of the study, as well as the confidentiality of data collection. All participants in the study gave informed consent, mostly with fingerprints and a few with written signatures. Participants could refrain from answering any questions and could withdraw their participation from the study at any time. All parts of the study were confidential, and all participant data were de-identified.

### 3.3. Data Collection

This study was conducted between March and September 2019 and employed a cross-sectional quantitative study design. After the ethical approval was received in March, the research team began to contact and work with two NGOs. The fieldwork for data collection was conducted from April to June among female MSWs, and from July to September among the male group. The research team included academics; government health staff from the Office of Disease Prevention and Control, Chiang Mai, who had been working with sex workers for a long time; and staff from two NGOs (M Plus Foundation and Perfect Life Foundation) who had direct experience providing sexual health services, outreach activities, and health promotion to sex workers in Chiang Mai.

Recruitment was divided into two steps. First, the research team organized meetings with partner NGOs to develop an agreement and clarify the background of the research, objectives, target group of the study, plan of data collection, and the importance of privacy and confidentiality. Local NGO staff developed a map of sex work venues to facilitate recruitment (18 sexual entertainment venues, 14 in urban and 4 in rural areas; 1 sexual health clinic, M Plus). Second, the staff from partner NGOs contacted and provided information about the study to the owners/managers of the venues. Most of the sex work venues welcomed the NGO staff since they had pre-existing good relationships connected to health promotion programs the partner NGOs provided to the venues for many years. NGO staff then invited everyone who met the inclusion criteria to join the study. Prospective participants were informed about the date and time that the research team would be at their workplaces to collect the data. Data collection was performed before their working hours. They were informed that participation in questionnaire survey was voluntary, and they could refrain from the study at any time. Monetary compensation (600 Thai Baht or USD 20) was provided for their time after the questionnaire had finished.

All participants met the study requirements. The sample of 396 participants (198 males and 198 females) was drawn from those who showed up at the date of the survey in each venue. The survey was administered face-to-face by gender-matched trained interviewers using a computer-based structured interview questionnaire (offline tablets). If the respondents felt more comfortable to speak in their own languages (i.e., Shan and Burmese), two members of the interview team were available to conduct interviews with them in their own languages. Participation in the interviews was anonymous. Apart from ID number, no names, signatures, or other identifying information were collected. The interviews took place in a room where participants felt comfortable and safe.

### 3.4. Variables and Categories

#### 3.4.1. Sociodemographics

Sociodemographic information collected included age (categorized as <20 years old, and then in increasing 5-year categories up to 49 years), place of birth (Thailand, Laos, Myanmar), ethnicity (Shan, Burmese, Laotian, or other), years of schooling in Thailand or elsewhere (<1 year, 2–5 years, 6–10 years, or more than 10 years), years of residency in Thailand (classified as <2 years, 2–5 years, 5–10 years, or more than 10 years), religion (Buddhist, Christian, Muslim, none), marital status (single, partnered, separated/divorced/widowed), number of children (0, 1, 2 or more), education (formal or non-formal education system, and then primary, secondary, high school, college/university, or other), and number of household members (classified as 1, 2–3, 4–5, or more than 5 persons). Place of employment was classified as karaoke venue, massage parlor, spa or sauna, restaurant, café, or road-side bar, and monthly income was recorded as <=5000, 5001–10,000, 10,001–15,000, 15,001–20,000, or ≥20,000 Baht. Questionnaires also collected information on identification documents such as a passport, a ‘pink card’ (an identification card for non-citizen residents), or a non-Thai identification card. Participants provided information on their possession of a work permit for Thailand, and their access to health insurance (categories were non-Thai resident health insurance, Thai 30-baht health care, private insurance, Thai Social Security Scheme).

#### 3.4.2. Sexual Behaviors

A variety of variables were measured regarding sexual behavior. These comprised age at first sexual intercourse (<15 years, 15–19 years, 20–24 years, >=25 years), identity of first sexual partner (boyfriend/girlfriend, friend, acquaintance, employer, stranger, client), number of sexual partners (>10, 10–49, 50–99, 100–149, 150–199, >=200 people), number of sexual partners per day in the past 3 months (none, 1–2, 3–4, >=5 persons), career duration (0–2 years, 2.1–5 years, 5.1–10 years, >10 years), main sexual behavior (vaginal sex, anal insertive, anal receptive, or oral sex), last sexual intercourse (1, 2–3, 4–5, 6–7, more than 7 days ago), most recent sexual partner (boyfriend/girlfriend, friend, acquaintance, employer, stranger, client), and protective methods during last sexual intercourse (none, withdrawal method, morning-after pill, condom, oral contraceptive pill, contraceptive implant).

### 3.5. Data Analysis

Prior to analyses, the data were screened for errors. Where errors occurred, the relevant interview was reviewed, and appropriate corrections were made. All variables were then visually inspected for missing data. In all categorical variables, missing data were re-labeled as ‘missing’ in the dataset. As the focus of this study was sociodemographic characteristics, and their association with condom use, derived variables were constructed based on common trends among sociodemographic characteristics. A dichotomous age variable was derived from the original continuous age variable. To aid interpretation, responses were grouped as under 30 years or over 30 years. An education variable was derived from original education responses. Responses were grouped by three education levels (‘no education’, referring to respondents who did not receive any level of schooling; ‘started primary school’, referring to respondents who received any education at the primary school level, whether completed or not; and ‘started high school’, referring to respondents who received any education at the high school level, whether completed or not). A condensed workplace variable was derived from the original six categories to include three categories (‘karaoke’, ‘eatery/bars’, ‘massage parlor’). The primary dependent variable was a composite variable for condom use with any partner (‘any partner’) derived from the dichotomous variables of client condom use, casual partner condom use, and regular partner condom use. Derived variables were used in place of the original variables in subsequent analyses, with the exceptions of client condom use (‘client partner’), casual partner condom use (‘casual partner’), and regular partner use (‘regular partner’); these were used in addition to the composite any partner variable. Sociodemographic variables of the participants were analyzed using frequency (%) and median and interquartile range, where appropriate. A series of bivariate chi-square tests were computed to assess the relationships between sociodemographic and sex behavior variables. Statistical significance was set at a 5% level for all analyses. Following bivariate analyses, condom use was analyzed by sociodemographic characteristics using logistic regression to assess the association between independent variables and consistent condom use. All analyses were performed using IBM SPSS Statistics (version 28.0.1.0, IBM, Armonk, NY, USA).

## 4. Results

### 4.1. Sociodemographic Characteristics

The age of the respondents ranged from 18 to 49 years old, with a median age of 25 (interquartile range [IQR] 22–30). Male respondents (median age 24, IQR 22–28) were significantly younger than females (median age 27, IQR 23–31; *p* = 0.003). Almost all respondents were Shan ethnicity (91.7%) and were born in Myanmar (92.9%). The majority of respondents had started primary school, with a higher proportion of female respondents starting high school (43.9%) than male respondents (34.5%). About half of them (51.9%) received basic education from their home country, while around 35.4% received education (at least primary school) in Thailand, divided into formal (53.6%) and non-formal (46.4%) education systems. A large number of respondents (41.2%) lived in Thailand for more than 10 years (average length of stay in Thailand = 11 years). Karaoke venues were the most common workplace among the whole sample population (51.0%), although there were differences in workplace between females and males. Most female migrant sex workers (FMSWs) worked in karaoke venues (84.3%), massage parlors (9.1%), and traditional Thai massage shops (8.6%), whereas male migrant sex workers (MMSWs) worked in a greater range of places, such as male spa pubs (39.4%), karaoke bars (20.7%), traditional Thai massage shops (17.7%), rural road-side bars (12.1%), restaurants (10.6%), and other entertainment complexes (10.1%).

Overall, 34.1% were in a committed relationship and 33.1% had children, and marriage status varied between male and female respondents. A greater proportion of MMSWs were single (64.1%) than FMSWs (47.5%). There were similar proportions of regular partners between MMSWs and FMSWs (32.8% and 35.4%, respectively).

### 4.2. Sexual Behavior and Condom Use

Sexual experience is also examined in Table 1. The median age at first sexual activity was 17 for both MMSWs and FMSWs (IQR 16–18). Career duration among participants ranged from 0 to 24 years. Two-thirds of all respondents (63.4%) had spent less than two years in sex work. This percentage was greater for MMSWs (69.7%) than FMSWs (57.1%). The majority of respondents (75.1%) had had sex with up to 99 different people (median 50, IQR 15–99), with 11.6% having had more than 200 lifetime sexual partners. One-third of FMSWs had 100 or more sexual partners compared to one-sixth of MMSWs. Among the whole sample population, condoms were the most common form of contraception reported for their last sexual intercourse (55.6%). The second highest was no contraception (25.8%), followed by the oral contraceptive pill (11.9%). FMSWs had higher reported rates of almost all contraceptive types compared to MMSWs, except for the withdrawal method. With respect to consistent condom use, only one-fifth of all respondents (19.4%) reported always using condoms in the past month regardless of partner. The rate of consistent condom use was marginally higher in FMSWs than MMSWs (20.6% and 18.5%, respectively); however, this difference was non-significant.

### 4.3. Sociodemographic Factors Affecting Condom Use

Frequency of condom use in the past month was split by education, age, sex, partner type, and workplace (see Table 2). Low rates (<25%) of consistent condom use were found across subgroups in age and workplace, suggesting that these did not significantly affect condom use. There was a marked difference in condom use by partner type: consistent condom use was practiced by 53.7% of respondents during sex with casual partners, and by 87.9% during sex with clients.

Chi-square analyses were then performed to examine the interaction between education, age, sex, partner type, or gender and condom use in the past month (see Table 2). Education was found to significantly affect condom use (χ^2^ = 8.08, *p* = *0*.018). Individual analyses revealed that consistent condom use was practiced significantly more frequently by respondents who started high school compared to respondents who started primary school (χ^2^ = 6.63, *p* = 0.01), but all other interactions between education level and condom use were non-significant. The relationship between sex partner type and condom use was also found to be significant (χ^2^ = 295.20, *p* < 0.001). Further analyses revealed significant interactions at the individual level. Consistent condom use was practiced more frequently during sex with casual partners compared to regular partners (χ^2^ = 66.01, *p* < 0.001), during sex with clients compared to regular partners (χ^2^ = 295.61, *p* < 0.001), and during sex with casual partners compared to clients (χ^2^ = 72.46, *p* < 0.001).

Figure 1 explores the relationships between education, age, workplace type, and partner type and consistent condom use, stratified by sex. There was a significant interaction between education and sex (χ^2^ = 8.98, *p* = 0.011). FMSWs with no education and FMSWs who started primary school reported higher rates of consistent condom use than MMSWs (26.2% vs. 6.4%, and 14.0% vs. 12.3%, respectively). Among respondents who started high school, MMSWs had higher rates of consistent condom use (30.0%) compared to FMSWs (22.0%). With respect to age, among participants under 30, a higher percentage of FMSWs practiced consistent condom use than MMSWs (21.5% vs. 17.0%). The opposite was true for respondents over 30 (25.8% and 18.6% for MMSWs and FMSWs, respectively). However, this relationship was non-significant (χ^2^ = 0.20, *p* = *0*.652). The relationship between workplace and sex was found to be significant (χ^2^ = 36.94, *p* < 0.001). FMSWs who worked in karaoke venues and eateries had higher rates of consistent condom use than MMSWs (30.0% vs. 8.6%, 100% vs. 18.1%, respectively). However, when considering workers in massage parlors, MMSWs had higher rates of consistent condom use than FMSWs (13.0%). A significant relationship was found between client type and sex (χ^2^ = 24.05, *p* < 0.001). FMSWs reported higher rates of consistent condom use across all partner types compared to MMSWs. Less than one-fifth of FMSWs consistently used condoms with regular partners (19.8%). For MMSWs, this decreased to 14.4%. During sex with casual partners, over half of both MMSWs and FMSWs used condoms consistently (52.1% and 59.5%, respectively). The majority of FMSWs and MMSWs used condoms during sex with clients (94.1% vs. 82.2%).

### 4.4. Predictors of Condom Use

Logistic regression models were constructed to examine predictors of condom use (see Table 3). Associations were measured between education, age, sex, and workplace and the likelihood of consistent condom use with regular partners, casual partners, clients, and with any partners. Logistic regression models were run for each individual predictor variable, and an adjusted model was run holding all included variables constant. While age was found to be non-significant in the bivariate analyses, the decision was made to retain age in the model, as it is an important sociodemographic variable.

Education was a significant predictor of consistent condom use with clients, regular partners, and with any partner. In relation to clients and regular partners, the likelihood of consistent condom use by respondents who started primary school was one-third the likelihood of respondents who started high school (AOR 0.34 [95% CI 0.13, 0.88], *p* = 0.027, and AOR 0.31 [95% CI 0.13, 0.73], *p* = 0.008, respectively). When predicting consistent condom use with any partner, respondents with no education were half as likely to always use condoms compared to respondents who started high school (AOR 0.46 [95% CI 0.22, 0.96], *p* = 0.038). Respondents who started primary school were only 0.40 times as likely to consistently use condoms with any partner compared to participants who started high school (AOR 0.40 [95% CI 0.19, 0.81] *p* = 0.011).

Sex was a significant predictor for consistent use of condoms with clients. MMSWs were one-quarter as likely to consistently use condoms during sex with clients compared to FMSWs (AOR 0.24 [95% CI 0.09, 0.63], *p* = 0.004). No other factors were found to significantly predict consistent condom use with clients.

Workplace was a significant predictor for consistent condom use with casual partners. Respondents who worked at a karaoke venue were one-fifth as likely to consistently use condoms with casual partners compared to respondents who worked in massage parlors (AOR 0.20 [95% CI 0.07], 0.56, *p* = 0.002).

Age was not a significant predictor for consistent condom use. 

## 5. Discussion

The findings of this study have helped to elucidate patterns of condom use by migrant sex workers in Chiang Mai. We found that less than one-fifth of MSWs consistently used condoms during sex with any partner (i.e., casual, regular, or client); however, this rate varied by partner type. Over half of MSWs always used condoms during sex with casual partners (53.7%). For sex with clients, this figure rose to 87.9%. While the cross-sectional nature of this study limits causal interpretability, our findings indicate that education and partner type were important factors in condom use among MSWs.

Similar to studies by Jirrattikorn et al., Chemnasiri et al., and Chamratrithirong and Kaiser, we found that sociodemographic factors, such as workplace and education, were correlated with consistent condom use [15,23,25]. MMSWs who worked in massage parlors had higher rates of consistent condom use than MMSWs in other workplaces. This could be due to greater promotion of condom use in massage parlors compared to other workplaces [26], although the low rate of condom use by FMSWs in massage parlors suggests otherwise. Respondents who started high school were most likely to consistently use condoms. Most surprisingly, the odds of consistently using condoms were greater among MSWs with no education, compared to MSWs who started primary school. We also found that FMSWs with no education had higher rates of consistent condom use than all other groups, aside from MMSWs who started high school. This differs from previous studies such as by Lagarde et al. [20] and Pinyopornpanish et al. [27], which found strong linear relationships between education and condom use. However, while we investigated school-based education, we did not investigate access to health education in adult life, which is a known predictor of condom use. This may explain why our findings conflict with previous studies. It may also be that associations between sociodemographic factors and consistent condom use were confounded by differences in sample size, although given the different trends found, further research is warranted.

The varying rates of condom use between sexual partners are in agreement with a plethora of research that found that condom use is higher with casual partners compared to regular partners [28,29,30]. In examining condom use and behavioral prediction, Rhodes et al. found that participants with casual partners had higher rates of condom use than men or women with regular partners, and this remained true even amongst controls. However, it is interesting that we found condom use was even greater with clients than non-commercial casual partners. It is possible that this is due to free condom distribution by the Thai government in the workplace [25].

In this study, we compared the condom use only between male and female MSWs. This was because the male MSWs in this study did not differentiate themselves as gay men or transgender women, even though they gave sexual services to all genders (e.g., gay men, transgender, heterosexual women). High rates of consistent condom use with clients were reported by both male and female sex workers. Notably, consistent condom use with clients among MSWs was higher than what was reported in previous studies that investigated Thai sex workers across Chiang Mai, Bangkok, and Phuket (TSWs) [22,23]. This is particularly interesting given the limited access to healthcare and health messaging for MSWs in Thailand, compared to TSWs [22,23]. For example, Barmania [24] reported on the discrepancy between MSWs and TSWs in healthcare access. While TSWs have free regular STD screening, no similar program exists for MSWs, meaning that testing costs are out of pocket. Similarly, health messaging for MSWs can be inaccessible, either due to language barriers or lack of opportunity [15,21,24]. Yet, despite this, our findings suggest that consistent condom use rates are in fact higher among migrant sex workers than Thai sex workers [22,23]. It may be that the Thai 100% condom campaign has increased acceptance of condom use among the client community, reducing this barrier. It is also possible that more sex workers are pushing for consistent condom use by clients. Jirattikorn et al. [15] found that some MSWs would only have sex with clients with condoms and would simply leave if clients refused. However, in the same study, other MSWs reported that clients would remove the condom during intercourse without consent. Further research on client attitudes and behavior relating to condom use is warranted.

The relatively low rate of consistent condom use by MSWs with regular and casual partners is some cause for concern. This may be due to alternative contraception being used with these partners or reduced perceived risk of STDs with non-client sexual partners. Nevertheless, it is still possible for transmission to occur during intercourse with non-clients, and the lack of consistent condom use increases this risk. Studies have found a positive correlation between social support and condom use among sex workers in China [31,32]; this warrants further exploration in Thai-based sex workers. In addition, further research on the disparity between condom use with regular and casual partners, compared to clients is needed.

We found that FMSWs had a higher rate of consistent condom use across sexual partners compared to MMSWs. Across research, there have been mixed findings regarding gender and condom use. Chamratrithirong and Kaiser [23] reported greater condom use among men, citing higher rates of sex with casual partners or sex workers as a cause. Conversely, Moraros et al. [33] found that adolescent female TSWs only used condoms 50% of the time. In the case of our findings, it is unclear what factors play a role in the higher rate of condom use among FMSWs. It is possible that consistent condom rates are higher among females in the MSW population than the TSW population, or it may be simply due to the more general risk of pregnancy for females. It would be useful to examine this more closely in further research.

### Limitations

There are several limitations to this study. The research was conducted in Chiang Mai and may not be generalizable to the other regions of Thailand. The focus of the study was migrant sex workers in Thailand, which means the findings may be less applicable to Thai sex workers. While the focus of the study on MSWs in Thailand may limit generalizability, we chose to address the paucity of research on this population. In addition, our study contributes to limited research specifically addressing the difficulties that MSWs face [7,8]. Our sample size was adequate for the overall study; however, power was limited for age and ethnicity, which limited our analyses. In addition, while there were equal sample sizes for males and females for the overall study, this was not the case for all sociodemographic factors, such as workplace, which may have affected analyses.

It is also possible that MSWs overreported consistent condom use with clients due to social desirability bias. Condom use is highly promoted in Thailand and is a particularly sensitive topic for sex workers. As questionnaires took place in the workplace, respondents may have felt safer to report higher condom use with clients to avoid conflict with their employers and co-workers, while being more honest about the condom use with partners in their personal life. In addition, questionnaires were administered by health personnel, which may have also contributed to overreporting of consistent condom use. It is also possible that respondents did not fully understand the questionnaire due to language barriers, which may have affected responses. That said, the respondents’ prior experience with the study health personnel and the low reported rates of consistent condom use with regular and casual partners indicate that respondents may have felt comfortable to answer honestly and ask for clarification if needed. This suggests that social desirability bias did not have a major effect on study outcomes.

## 6. Conclusions

With this study, we sought to contribute to understandings regarding sociodemographic factors and condom use among the neglected population of MSWs. We found that sociodemographic factors were associated with condom use among MSWs in Chiang Mai, especially with sexual partner type and education. We identified higher rates of consistent condom use by migrant sex workers during sex with clients compared to other studies, particularly among female migrant sex workers. However, the overall consistent condom rate remained low for both male and female migrant sex workers. These findings indicate the need for further health education for MSWs, particularly regarding consistent condom use across sexual partners. More broadly, future research should address the role that client preferences play in consistent condom use, and how this may be affected by the use of other contraceptives. The important role than gender norms play in consistent condom use by sex workers, and social norms among other sex workers and friends have been examined in public health literature, but further exploration of condom use among FMSWs compared to MMSWs is warranted. Finally, while the differences in condom use between regular and casual partners are known, it would be useful to examine the factors affecting client condom use and condom use with non-commercial partners.

## Figures and Tables

**Figure 1 ijerph-19-09830-f001:**
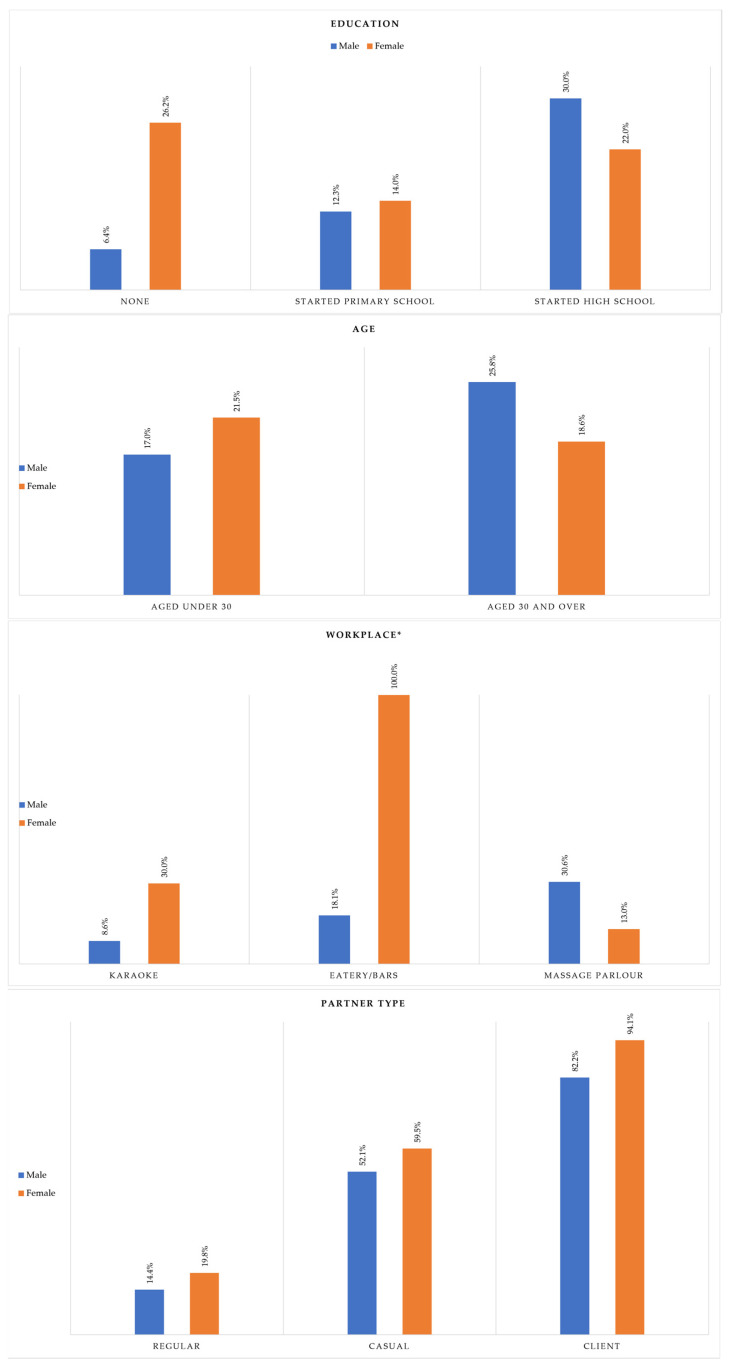
Relationships between consistent condom use and education, age, workplace, and partner, stratified by sex. * Note that sample size for FMSWs in eatery/bars was 1.

**Table 1 ijerph-19-09830-t001:** Demographics of sample population.

Demographics	Male (*n* = 198) No. (%)	Female (*n* = 198) No. (%)	Total (*n* = 396) No. (%)	*p*-Value ^1^
Age				<0.001
<30	164 (82.8%)	133 (67.2%)	297 (75.0%)	
30+	34 (17.2%)	65 (32.8%)	99 (25.0%)	
Median (IQR)	24 (22–28)	27 (23–31)	35 (22–30)	
Range	15–49	18–49	15–49	
Race/ethnicity ^2^				0.025
Shan/Tai-Yai	170 (92.9%)	171 (90.5%)	341 (91.7%)	
Burmese	9 (4.9%)	3 (1.6%)	12 (3.2%)	
Laotian	1 (0.5%)	3 (1.6%)	4 (1.1%)	
Other	3 (1.6%)	12 (6.3%)	15 (4.0%)	
Education in any country ^3^				0.073
No schooling	48 (24.5%)	67 (34.0%)	115 (29.3%)	
Primary school or less	31.6%)	62 (31.5%)	124 (31.6%)	
Started high school	86 (43.9%)	68 (34.5%)	154 (39.2%)	
Workplace ^4^				<0.001
Karaoke	37 (19.7%)	160 (80.8%)	197 (51.0%)	
Eatery/bar ^^^	113 (60.1%)	5 (2.5%)	118 (30.6%)	
Massage ^^^^	38 (20.2%)	33 (16.7%)	71 (18.4%)	
Marital status				<0.001
Single	127 (64.1%)	94 (47.5%)	221 (55.8%)	
Partnered	65 (32.8%)	70 (35.4%)	135 (34.1%)	
Separated/divorced	6 (3.0%)	34 (17.2%)	40 (10.1%)	
Age at first sexual experience				0.043
<15 years	30 (15.2%)	14 (7.1%)	44 (11.1%)	
15–19 years	143 (72.2%)	153 (77.3%)	296 (74.7%)	
20–24 years	24 (12.1%)	27 (13.6%)	51 (12.9%)	
>=25 years	1 (0.5%)	4 (2.0%)	5 (1.3%)	
Median (IQR)	17 (16–18)	17 (16–18)	17 (16–18)	
Range	8–25	8–28	8–28	
Career duration				0.040
1. 0–2 years	138 (69.7%)	113 (57.1%)	251 (63.4%)	
2. 2.1–5 years	41 (20.7%)	57 (28.8%)	98 (24.7%)	
3. 5.1–10 years	17 (8.6%)	21 (10.6%)	38 (9.6%)	
4. >10 years	2 (1.0%)	7 (3.5%)	9 (2.3%)	
Median (IQR)	1 (0–3)	2 (0–4)	1 (0–3)	
Range	0–17	0–24	0–24	
No. sexual partners in lifetime				<0.001
<10	29 (14.6%)	37 (18.7%)	66 (16.7%)	
10–49	69 (34.8%)	56 (28.3%)	125 (31.6%)	
50–99	67 (33.8%)	39 (19.7%)	106 (26.8%)	
100–149	13 (6.6%)	31 (15.7%)	44 (11.1%)	
150–199	2 (1%)	7 (3.5%)	9 (2.3%)	
>200	18 (9.1%)	28 (14.1%)	46 (11.6%)	
Median (IQR)	50 (15–60)	50 (10–100)	50 (15–99)	
Range	1–3000	1–2592	1–3000	
Contraception during last sexual intercourse				<0.001
None	73 (36.9%)	29 (14.6%)	102 (25.8%)	
Withdrawal method	13 (6.6%)	5 (2.5%)	18 (4.5%)	
Morning-after pill	1 (0.5%)	2 (1.0%)	3 (0.8%)	
Condom	92 (46.5%)	128 (64.6%)	220 (55.6%)	
Oral contraceptive pill	18 (9.1%)	29 (14.6%)	47 (11.9%)	
Contraceptive implant	1 (0.5%)	5 (2.5%)	6 (1.5%)	
Condom use in past month ^5^				0.637
Always	34 (18.5%)	28 (20.6%)	62 (19.4%)	
Sometimes or never	150 (81.5%)	108 (79.4%)	258 (80.6%)	

Note. ^1^ A bivariate analysis was performed using a chi-square test to determine if there was a statistically significant association between sociodemographic factors and sex. ^2^ Male *n* = 183, female *n* = 189. ^3^ Male *n* = 196, female *n* = 197. ^4^ Male *n* = 188, female *n* = 198. ^5^ Male *n* = 184, female *n* = 136. ^^^ Eateries and bars include restaurants, cafes, rural road-side bars, pubs, and bars. ^^^^ Massage parlors include traditional massage, spa and sauna, and massage parlors.

**Table 2 ijerph-19-09830-t002:** Frequency of condom use in the past month by education, age, sex, and workplace. Demographics.

	Always	*p*-Value
Education in any country		0.018
No schooling (*n* = 89)	14(15.7%)	
Started primary school (*n* = 100)	13 (13.0%)	
Started high school (*n* = 130)	35 (26.9%)	
Age		0.577
<30 (*n* = 246)	46 (18.7%)	
30+ (*n* = 74)	16 (21.6%)	
Sex		0.637
Male (*n* = 184)	34 (18.5%)	
Female (*n* = 136)	28 (20.6%)	
Partner type		<0.001
Regular partner (*n* = 265)	45 (17.0%)	
Casual partner (*n* = 177)	95 (53.7%)	
Client (*n* = 321)	282 (87.9%)	
Workplace		0.665
Karaoke (*n* = 147)	27 (18.4%)	
Pub/bar/restaurant (*n* = 106)	20 (18.9%)	
Massage parlor (*n* = 59)	14 (23.7%)	
Marital status		0.048
Single (*n* = 162)	37 (22.8%)	
Partnered (*n* = 130)	17 (15.0%)	
Separated/divorced/widowed (*n* = 28)	8 (28.6%)	
Career duration		0.491
1. 0–2 years (*n* = 205)	38 (18.5%)	
2. 2.1–5 years (*n* = 79)	14 (17.7%)	
3. 5.1–10 years (*n* = 31)	8 (25.8%)	
4. >10 years (*n* = 5)	2 (40.0%)	
Total (*n* = 320)	62 (19.4%)	

**Table 3 ijerph-19-09830-t003:** Multivariate logistic regression of consistent condom use by education, age, sex, and workplace (stratified by partner type) ^1^.

	Client Partner				Casual Partner				Regular Partner				Any Partner			
	OR [95% CIs]	*p*-Value	AOR [95% CIs]	*p*-Value	OR [95% CIs]	*p*-Value	AOR [95% CIs]	*p*-Value	OR [95% CIs]	*p*-Value	AOR [95% CIs]	*p*-Value	OR [95% CIs]	*p*-Value	AOR [95% CIs]	*p*-Value
Education ^2^																
None	0.66 [0.27, 1.65]	0.38	0.47 [0.17, 1.32]	0.152	0.52 [0.26, 1.06]	0.073	0.54 [0.23, 1.24]	0.145	0.61 [0.29, 1.32]	0.211	0.57 [0.25, 1.29]	0.175	0.51 [0.25, 1.01]	0.053	0.46 [0.22, 0.96]	0.038
Started primary school	0.43 [0.18, 1.01]	0.052	0.34 [0.13, 0.88]	0.027	0.95 [0.45, 1.99]	0.89	0.98 [0.44, 2.15]	0.957	0.31 [0.13, 0.73]	0.008	0.31 [0.13, 0.73]	0.008	0.41 [0.20, 0.82]	0.011	0.40 [0.19, 0.81]	0.011
Age ^3^																
<30	0.54 [0.23, 1.28]	0.160	0.74 [0.30, 1.87]	0.527	0.69 [0.35, 1.40]	0.306	0.62 [0.27, 1.41]	0.253	1.00 [0.46, 2.17]	0.994	0.91 [0.40, 2.11]	0.833	0.83 [0.44, 1.58]	0.577	0.80 [0.40, 1.62]	0.543
Sex ^4^																
Male	0.29 [0.13, 0.64]	0.002	0.24 [0.09, 0.63]	0.004	0.74 [0.36, 1.55]	0.428	0.55 [0.21, 1.43]	0.219	0.68 [0.36, 1.29]	0.239	0.81 [0.31, 2.07]	0.654	0.87 [0.50, 1.53]	0.637	0.76 [0.34, 1.70]	0.498
Workplace ^4^																
Karaoke	1.11 [0.43, 2.89]	0.833	0.74 0.26, 2.12]	0.58	0.21 [0.08, 0.54]	0.001	0.20 [0.07, 0.56]	0.002	1.78 [0.69, 4.63]	0.235	1.55 [0.55, 4.40]	0.412	0.72 [0.35, 1.50]	0.385	0.64 [0.28, 1.44]	0.283
Eatery/bars	0.71 [0.27, 1.86]	0.484	1.33 [0.47, 3.76]	0.592	0.36 [0.14, 0.88]	0.026	0.44 [0.16, 1.03]	0.059	1.24 [0.43, 3.54]	0.431	1.20 [0.39, 3.77]	0.750	0.75 [0.35, 1.62]	0.460	0.77 [0.33, 1.81]	0.554

Note. ^1^ In the adjusted models, education, age, sex, and workplace were all included. ^2^ Reference category was started high school. ^3^ Reference category was ≥30. ^4^ Reference category was massage parlors.

## Data Availability

The data presented in this study are openly available in FigShare at 10.6084/m9.figshare.19367189 (accessed on 16 March 2022).

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
