# Peer review of "Association between Sociodemographic Factors and Condom Use among Migrant Sex Workers in Chiang Mai, Northern Thailand"

_ijerph, 2022, doi:10.3390/ijerph19169830_

Round 1
Reviewer 1 Report
Thank you for the opportunity to review this revised paper. While the authors did some work to improve the paper, it is not ready for publication, and below I reiterate many of my earlier comments, which were meant to help the authors clarify their argument and the body of scholarship they are contributing to, and how they are making this contribution.
In my initial review, I asked “What is the paper’s main argument?” In this revised paper, there is now an intro with some vague statements about global HIV and condom use findings, but there is still no clearly stated question and argument specific to this paper. The closest the authors come to stating a question is now at lines 97-101. This material should be earlier, in the intro, with a clearly stated overall argument for the paper—ie, as I asked in my previous review, what is the broader argument about condom use among MSWs that we can take from these individual statistics?? It seems that this argument does not appear until line 424: “We found that sociodemographic factors were associated with condom use among MSWs in Chiang Mai, espe cially with sexual partner type and education.” Move this up!
While the paper now includes a background section outlining the prevalence of HIV in Thailand, there still lacks any broader theoretical or research framing. As I requested before, “What theory/broader literature informs this study? For example, public health literature on HIV/AIDS prevention? To develop this, the authors should create a lit review section that precedes the materials and methods section. This lit review should clearly state the gap in research that their project fills.” Without this material added to the paper, it remains unclear what research gap (or, research need) this paper, on this specific topic, is filling.
The authors now seem to explain a bit more (in material that was largely moved and re-organized into the new background section) why they focus on MSWs. This is helpful.
In the previous review, I asked “Explain more the time-frame (March -Sept 2019).” While the authors now provide some detail about what they DID in this timeframe (lines 120+), this is not the same as explaining WHY this time period is relevant for study—for example, what sparked them to start this study in that time period?
In the draft I received, the authors have still not addressed the issues I raised re. their survey (or, I could not see where they did this). Therefore, I will state this request again, from my last review: “The survey instrument also needs more explanation. Since there is no literature review, it is difficult to discern in the paper from where the survey questions and topics (outlined in section 2.4) were derived, i.e. we don’t know why they are surveying on these topics. I don’t think that it is enough to just say that a study is using “validated instruments”—why are these being used? As well, explain more why the survey was analyzed in the way it was (section 2.5) – to what end? How does this help to answer the research question (which, again, needs to be stated more clearly at the outset of the paper).
It also does not seem like the authors did anything to revise the findings section, and it would be at least interesting to know why. Without any visible work here, I will ask the following again, “The findings section reads as a list of statistical findings which, while interesting, are not presented in a way that supports a particular argument (see earlier comment that the paper lacks a clear research question and response, i.e. the over-arching argument). What are we learning about migrant sex workers from these statistics about their age, education level etc? What broader argument is all of this data advancing about condom use among MSW populations?”
The authors do discuss the education component more, which is interesting, but it is only clear how it relates to/supports their main argument around line 424, which is why the argument needs to come earlier, and then the findings need to be more clearly presented to illustrate and support this argument.
The authors did a bit of work revising the conclusion, and they note their contribution to a broader body of lit (“public health literature”). But since this literature is not reviewed anywhere, coherently and earlier on in the paper (to show a broader gap in research), it is unclear what public health literature they are advancing and how. Also, they could say more about directions for future research, to elaborate the one line they added at 437-38.
Author Response
Please see the attachment.

This manuscript is a resubmission of an earlier submission. The following is a list of the peer review reports and author responses from that submission.
Round 1
Reviewer 1 Report
Thank you for the opportunity to review “Sexual Partners and Predictors of Condom Use”. This paper presents data from a very interesting study of condom use by migrant sex workers in Thailand. As it is currently written, the paper is not ready for publication, and below I discuss how the authors may revise it to better clarify their central argument, population focus, findings, and broader body of literature with which they engage.
Argument
What is the paper’s main argument? The abstract mainly lists a various findings, but this does not indicate what is the paper’s overall argument (and purpose). The paper’s main argument is also not presented clearly (or at all) in the introduction. While the intro provides background about the project, it never clearly states a central research question and over-arching argument (i.e. a response to this question). I therefore suggest that the authors make the intro more concise to explain their focus on MSWs (see comment below), clearly state their research question, and provide a clear statement of their argument.
To develop this argument, I suggest that the authors revisit this statement: “Associations between condom use and sociodemographic factors were examined using chi-square tests and 22 logistic regression models. A significant association was found between condom use and regular 23 partners, (χ2=295.20, p<.001) and between condom use and starting high school (χ2=8.08, p=.018). 24 Education was the only variable that was a significant predictor of condom use with any sexual 25 partner (OR=0.41; 95%CI 0.20-0.82).”
What is the broader argument about condom use among MSWs that we can take from these individual statistics?
Literature and context
This paper lacks a clear literature review that situates the project within a broader scholarly discussion or theoretical frame. What theory/broader literature informs this study? For example, public health literature on HIV/AIDS prevention? To develop this, the authors should create a lit review/background section that precedes the materials and methods section. This lit review should clearly state the gap in research that their project fills. Providing this literature review earlier in the paper is important bc the authors write later (line 318): “Similar to other studies, we found that sociodemographic factors, such as workplace 318 and education, were correlated with consistent condom use [13, 17, 18].” These studies (and many others referenced in the discussion) need to be reviewed earlier, so the reader knows from what body of research this particular study is derived, and to what it is contributing.
It would also be helpful for the authors to situate their project in a broader context, and discuss at least briefly at the outset why the AP region maintains such stubbornly high HIV rates, despite its plans to eradicate this virus by 2030. Is this a result of policy failure? Economics (at least in the US, poverty is a significant factor in vulnerability to HIV /AIDS)? Other? Providing this information would help to contextualize the research and indicate its importance. Much of the material in what is currently the intro could then go in this contextual discussion too.
Explaining the focus: (M)SWs
The authors most justify more their focus on SWs, and MSWs in particular. As they write at line 46, “Of all new infections, around 44% occurred among men who have sex with men and 10% among sex workers and their clients [6]. These two groups are also over-represented in terms of prevalence with around 2% of Thailand’s 147,000 sex workers and 12% of men who have sex with men currently infected with HIV [2, 7].” From this information, it seems to me that if we should be concerned about and research about any population’s condom use, it should be men who have sex with men (MSM), not sex workers, as MSMs have more new infections than other populations (incl SWs) by a long shot! Furthermore, even among MSWs, HIV prevalence is lower for them (5%) than it is for MSMs (12%), if I am reading these number correctly. So, all of this is to say that the authors need to explain much more clearly and explicitly why they are focused on an interested in condom use among SWs (and MSWs in particular), and not MSMs, so that they do not reify the long-held stereotype that SWs (and even MSWs) are the most at risk “vector of disease” population, despite evidence (that they present) to the contrary.
Materials and methods
Explain more the time-frame (March -Sept 2019).
It is also imperative that the research team position itself more—do they have sex work experience? Language proficiency? Relationships with the community studied?
Who were the NGO partners (What do they do?)? And why the focus on Chiang Mai?
The survey instrument also needs more explanation. Since there is no literature review, it is difficult to discern in the paper from where the survey questions and topics (outlined in section 2.4) were derived, i.e. we don’t know why they are surveying on these topics. I don’t think that it is enough to just say that a study is using “validated instruments”—why are these being used? As well, explain more why the survey was analyzed in the way it was (section 2.5) – to what end? How does this help to answer the research question (which, again, needs to be stated more clearly at the outset of the paper).
What, then, did the interviews cover, and how was this data analyzed? Did all of the interviewees complete the interviews too, or were interviews only with a subset of survey respondents?
Findings
The findings section reads as a list of statistical findings which, while interesting, are not presented in a way that supports a particular argument (see earlier comment that the paper lacks a clear research question and response, i.e. the over-arching argument). What are we learning about migrant sex workers from these statistics about their age, education level etc? What broader argument is all of this data advancing about condom use among MSW populations? It seems that education is something important here-- maybe consider revising and presenting the findings more to focus on/emphasize a potential argument about this?
I am also curious to hear more in the paper why the authors compared sex workers by gender, and why their gender breakdown was so binaristic—did all MSWs identify as men or women only? Were there no transgender or other non-binary sex workers? Why not?
Also, how do their findings compare to sex workers in other parts of Thailand? While did they not do a comparative study, as noted in the limitations, surely there is data about condom use in the general Thai SW population that they could discuss.
The conclusion also needs much more development. How does their research advance a broader body of literature? I suggest that they re-organize and move material from the parts of the discussion section etc to create a conclusion that does the following: 1) indicates 2-3 contributions from this research to the broader literature/theory this project engages (see earlier comments); 2) discusses 2-3 contributions this research makes to more specific topics such as sex work in Thailand; and 3) directions for future research.
Reviewer 2 Report
This paper covers an interesting topic. The methods are good.
One thing to re-think is you have the word ‘predictor’ in your title. But your study is cross-sectional. Predictors are for prospective cohort studies. Have a think if this word stays in the title.
The issue is the writing style. I got half-way through the methods and it became exhausting and I gave up. The paper needs a thorough edit. There are some senior authors from ANU on the paper who have clearly not read this paper but allowed it to be submitted. I am not an author and wont edit it for you. This is your job.
Abstract
line: 14.
Grammar, Information on the sexual ….
Line 18: ‘were assessed’. How? Using what?
U can just add a few words to the end of this sentence to answer this.
line 22: you have provided results from lines 18-21 and now you have returned to providing more methods. Put all the methods together and then all the results together. Move line 22 to line 18.
Line 23: having looked at table 3, I think you should say ‘consistent condom use’.
Line: 24. You can just provide the p-value here as these χ2 are not in your tables.
Line: 27. ‘proper’ I think your mean ‘better’
Background
Line 32: this is clunky English. This sentence should be… The Asia-Pacific region has the second largest…
Line 35: The incidence of HIV in Thailand peaked in the 1990s…
Line 38: delete the word Moreover,
Line 58: delete the word ‘cases’
Lines 65-67: this sentence is wordy. Cut the one-sixth statement and just say an estimated 200,000 of the 1.2 million population are …
Line 73: re-write this last sentence. The problem is ‘at up to’.
Line 85: delete ‘the left behind group of’
Lines 78-85: Your study aims to look at condom use and STIs among MSWs. So it does not fill the gap of the UNAIDS 90-90-90 goal and it does not fill the gap in access to HIV testing because its only for Thai citizens. So you really need to re-write this paragraph. You are just doing a cross sectional cut of condom use. this paragraph makes two large over statements. You should just write that this study will provide information about condom use among MSW so services could be designed to meet their needs.
Methods
Line 88: This first bit should go under the heading ‘Data collection’. The sentence should start with, A cross-sectional study design was used.
Line 92: this is wordy again. It should just be. The eligibility criteria were: (1) …
Line 104: delete the word ‘methods’
Line 105: delete ‘for participation in the study’
Lines 108-109: this sentence should just start with ‘Second, the staff from the ….’
Wordy writing/ short clear sentences will always get you further.
Line 111: delete ‘identified’
Line 113: this sentence doesn’t make sense. Re-write. All inclusion criteria???
Line 114: grammar
Line 119-120: re-think how you have written his sentence. It is very chunky.
Lines 121-122: you have just done it again. There are 19 recruitment sites…
There are too many grammatical and clunky English writing mistakes for me to continue.
Round 2
Reviewer 1 Report
Thank you for the opportunity to review this revised paper. I think the authors have done some work to improve the paper. Below I suggest how they authors may revise the paper to provide and introductory section with a clear statement of the argument, and I discuss how I still think the authors could do more work to situate their paper more clearly and explicitly in a broader body of literature (beyond the very area specific material they now review) so that the paper is more appealing and interesting to the journal’s broad readership.
The focus on and justification for MSWs is much clearer now, given the additions to line 182. However, the paper still lacks, by my reading, a clear—and SHORT/CONCISE (ie no more than 1 double spaced page)—introductory section that states the paper’s central problem, question, argument, and roadmap for the paper.
While the authors did revise lines 5-30 to state a broader argument to some extent, this is in the abstract, not the intro, and the statement of the broader argument in the abstract is not as clear here as it is in their response memo, where they write, “The broader argument regarding condom use among MSWs is that condom use was most consistent during sex with clients, when compared with other sexual partners, and that higher education (at least some high school) was positively correlated with condom use”—a version of THIS needs to be in the intro (actually, the main argument is now stated most clearly in the revised conclusion).
The intro should be followed by a lit review (outlining a broader area of scholarship beyond that in the specific geographic region) that states the gap in research, then a detailed justification of their case (ie much of the material that is currently in the intro), followed by the methods to show how they studied this case.
Overall, I still think the literature review (currently presented as an intro) remains narrow—while it focuses on explaining their focus on MSWs in a particular region, it is still not clear what research gap they are filling more broadly-- eg in public health research/theory more broadly. While it is true that they now show research on an under-studied impacted population, what lessons does this offer to scholars who may not care about the region but be interested in HIV/AIDS prevention, public health, etc? Again, as I noted before, they write in the discussion, “Similar to other studies….” What other studies? What do they say? These studies need to be reviewed as part of a broader lit review section, at the outset of the paper (following the intro), to indicate a the gap in research they fill.
The methods section is now clearer, and the added detail is useful.
I still would rather see future research in the conclusion, and I also think the authors could do more here, or even in the discussion, to indicate the lessons this paper offers scholars in broader fields (eg public health), but I will leave it up to the editor to make this decision.